# The Singular Values of Convolutional Layers

**Hanie Sedghi, Vineet Gupta and Philip M. Long**
Google Brain
Mountain View, CA 94043
{hsedghi,vineet,plong}@google.com

## Abstract

We characterize the singular values of the linear transformation associated with a standard 2D multi-channel convolutional layer, enabling their efficient computation. This characterization also leads to an algorithm for projecting a convolutional layer onto an operator-norm ball. We show that this is an effective regularizer; for example, it improves the test error of a deep residual network using batch normalization on CIFAR-10 from 6.2% to 5.3%.

## 1 Introduction

Exploding and vanishing gradients (Hochreiter, 1991; Hochreiter et al., 2001; Goodfellow et al., 2016) are fundamental obstacles to effective training of deep neural networks. Many deep networks used in practice are layered. We can think of such networks as the composition of a number of feature transformations, followed by a linear classifier on the final layer of features. The singular values of the Jacobian of a layer bound the factor by which it increases or decreases the norm of the backpropagated signal. If these singular values are all close to 1, then gradients neither explode nor vanish. These singular values also bound these factors in the forward direction, which affects the stability of computations, including whether the network produces the dreaded "Nan". Moreover, it has been proven (Bartlett et al., 2017) that the generalization error for a network is bounded by the Lipschitz constant of the network, which in turn can be bounded by the product of the operator norms of the Jacobians of the layers. Cisse et al. (2017) discussed robustness to adversarial examples as a result of bounding the operator norm.

These considerations have led authors to regularize networks by driving down the operator norms of network layers (Drucker and Le Cun, 1992; Hein and Andriushchenko, 2017; Yoshida and Miyato, 2017; Miyato et al., 2018). Orthogonal initialization (Saxe et al., 2013; Pennington et al., 2017) and Parseval networks (Cisse et al., 2017) are motivated by similar considerations.

Convolutional layers (LeCun et al., 1998) are key components of modern deep networks. They compute linear transformations of their inputs. The Jacobian of a linear transformation is always equal to the linear transformation itself. Because of the central importance of convolutional layers to the practice of deep learning, and the fact that the singular values of the linear transformation computed by a convolutional layer are the key to its contribution to exploding and vanishing gradients, we study these singular values. Up until now, authors seeking to control the operator norm of convolutional layers have resorted to approximations (Yoshida and Miyato, 2017; Miyato et al., 2018; Gouk et al., 2018a). In this paper, we provide an efficient way to compute the singular values exactly — this opens the door to various regularizers.

We consider the convolutional layers commonly applied to image analysis tasks. The input to a typical layer is a feature map, with multiple channels for each position in an $n \times n$ field. If there are $m$ channels, then the input as a whole is a $m \times n \times n$ tensor. The output is also an $n \times n$ field with multiple channels per position[1]. Each channel of the output is obtained by taking a linear combination of the values of the features in all channels in a local neighborhood centered at the corresponding position in the input feature map. Crucially, the same linear combination is used for all positions in the feature map. The coefficients are compiled in the *kernel* of the convolution. If the neighborhood is a $k \times k$ region, a kernel $K$ is a $k \times k \times m \times m$ tensor. The projection $K_{:,:,c,:}$ gives the coefficients

---

[1]Here, to keep things simple, we are concentrating on the case that the stride is 1.

that determine the $c$th channel of the output, in terms of the values found in all of the channels of all positions in it neighborhood; $K_{:,:,c,d}$ gives the coefficients to apply to the $d$th input channel, and $K_{p,q,c,d}$ is the coefficient to apply to this input at in the position in the field offset horizontally by $p$ and vertically by $q$. For ease of exposition, we assume that feature maps and local neighborhoods are square and that the number of channels in the output is equal to the number of channels in the input - the extension to the general case is completely straightforward.

To handle edge cases in which the offsets call for inputs that are off the feature maps, practical convolutional layers either do not compute outputs (reducing the size of the feature map), or pad the input with zeros. The behavior of these layers can be approximated by a layer that treats the input as if it were a torus; when the offset calls for a pixel that is off the right end of the image, the layer "wraps around" to take it from the left edge, and similarly for the other edges. The quality of this approximation has been heavily analyzed in the case of one-dimensional signals (Gray, 2006). Consequently, theoretical analysis of convolutions that wrap around has been become standard. This is the case analyzed in this paper.

**Summary of Results:** Our main result is a characterization of the singular values of a convolutional layer in terms of the kernel tensor $K$. Our characterization enables these singular values to be computed exactly in a simple and practically fast way, using $O(n^2m^2(m + \log n))$ time. For comparison, the brute force solution that performs SVD on the matrix that encodes the convolutional layer's linear transformation would take $O((n^2m)^3) = O(n^6m^3)$ time, and is impractical for commonly used network sizes. As another point of comparison, simply to compute the convolution takes $O(n^2m^2k^2)$ time. We prove that the following two lines of NumPy correctly compute the singular values.

```
def SingularValues(kernel, input_shape):
  transforms = np.fft.fft2(kernel, input_shape, axes=[0, 1])
  return np.linalg.svd(transforms, compute_uv=False)
```

Here `kernel` is any $k \times k \times m \times m$ tensor[2] and `input_shape` is the shape of the feature map to be convolved. A TensorFlow implementation is similarly simple.

Timing tests, reported in Section 4.1, confirm that this characterization speeds up the computation of singular values by multiple orders of magnitude – making it usable in practice. The algorithm first performs $m^2$ FFTs, and then it performs $n^2$ SVDs. The FFTs, and then the SVDs, may be executed in parallel. Our TensorFlow implementation runs a lot faster than the NumPy implementation (see Figure 1); we think that this parallelism is the cause. We used our code to compute the singular values of the convolutional layers of the official ResNet-v2 model released with TensorFlow (He et al., 2016). The results are described in Appendix C.

Exposing the singular values of a convolutional layer opens the door to a variety of regularizers for these layers, including operator-norm regularizers. In Section 4.2, we evaluate an algorithm that periodically projects each convolutional layer onto a operator-norm ball. Using the projections improves the test error from 6.2% to 5.3% on CIFAR-10. We evaluate bounding the operator norm with and without batchnorm and we see that regularizing the operator norm helps, even in the presence of batch normalization. Moreover, operator-norm regularization and batch normalization are not redundant, and neither dominates the other. They complement each other.

**Related work:** Prior to our work, authors have responded to the difficulty of computing the singular values of convolutional layers in various ways. Cisse et al. (2017) constrained the matrix to have orthogonal rows and scale the output of each layer by a factor of $(2k+1)^{-\frac{1}{2}}$, for $k \times k$ kernels. Gouk et al. (2018a;b) proposed regularizing using a per-mini-batch approximation to the operator norm. They find the largest ratio between the input and output of a layer in the minibatch, and then scale down the transformation (thereby scaling down all of the singular values, not just the largest ones) so that the new value of this ratio obeys a constraint.

Yoshida and Miyato (2017) used an approximation of the operator norm of a reshaping of $K$ in place of the operator norm for the linear transformation associated with $K$ in their experiments. They reshape the given $k \times k \times m \times m$ into a $mk^2 \times m$ matrix, and compute its *largest* singular value using a power iteration method, and use this as a substitute for the operator norm. While this provides

---

[2]The same code also works if the filter height is different from the filter width, and if the number of channels in the input is different from the number of channels of the output.

a useful heuristic for regularization, the largest singular value of the reshaped matrix is often quite different from the operator norm of the linear transform associated with $K$. Furthermore if we want to regularize using projection onto an operator-norm ball, we need the whole spectrum of the linear transformation (see Section 3). The reshaped $K$ has only $m$ singular values, whereas the linear transformation has $mn^2$ singular values of which $mn^2/2$ are distinct except in rare degenerate cases. It is possible to project the reshaped $K$ onto an operator-norm ball by taking its SVD and clipping its singular values — we conducted experiments with this projection and report the results in Section 4.4.

A close relative of our main result was independently discovered by Bibi et al. (2019, Lemma 2).

**Overview of the Analysis:**    If the signal is 1D and there is a single input and output channel, then the linear transformation associated with a convolution is encoded by a *circulant matrix*, i.e., a matrix whose rows are circular shifts of a single row (Gray, 2006). For example, for a row $a = (a_1, a_2, a_3)$, the circulant matrix $\mathrm{circ}(a)$ generated by $a$ is $\begin{pmatrix} a_0 & a_1 & a_2 \\ a_2 & a_0 & a_1 \\ a_1 & a_2 & a_0 \end{pmatrix}$. In the special case of a 2D signal with a single input channel and single output channel, the linear transformation is *doubly block circulant* (see (Goodfellow et al., 2016)). Such a matrix is made up of a circulant matrix of blocks, each of which in turn is itself circulant. Finally, when there are $m$ input channels and $m$ output channels, there are three levels to the hierarchy: there is a $m \times m$ matrix of blocks, each of which is doubly block circulant. Our analysis extends tools from the literature built for circulant (Horn and Johnson, 2012) and doubly circulant (Chao, 1974) matrices to analyze the matrices with a third level in the hierarchy arising from the convolutional layers used in deep learning. One key point is that the eigenvectors of a circulant matrix are Fourier basis vectors: in the 2D, one-channel case, the matrix whose columns are the eigenvectors is $F \otimes F$, for the matrix $F$ whose columns form the Fourier basis. Multiplying by this matrix is a 2D Fourier transform. In the multi-channel case, we show that the singular values can be computed by (a) finding the eigenvalues of each of the $m^2$ doubly circulant matrices (of dimensions $n^2 \times n^2$) using a 2D Fourier transform, (b) by forming multiple $m \times m$ matrices, one for each eigenvalue, by picking out the $i$-th eigenvalue of each of the $n^2 \times n^2$ blocks, for $i \in [1..n^2]$. The union of all of the singular values of all of those $m \times m$ matrices is the multiset of singular values of the layer.

**Notation:**    We use upper case letters for matrices, lower case for vectors. For matrix $M$, $M_{i,:}$ represents the $i$-th row and $M_{:,j}$ represents the $j$-th column; we will also use the analogous notation for higher-order tensors. The operator norm of $M$ is denoted by $||M||_2$. For $n \in \mathbb{N}$, we use $[n]$ to denote the set $\{0, 1, \ldots, n-1\}$ (instead of usual $\{1, \ldots, n\}$). We will index the rows and columns of matrices using elements of $[n]$, i.e. numbering from 0. Addition of row and column indices will be done mod $n$ unless otherwise indicated. (Tensors will be treated analogously.) Let $\sigma(\cdot)$ be the mapping from a matrix to (the multiset of) its singular values. [3]

Let $\omega = \exp(2\pi i/n)$, where $i = \sqrt{-1}$. (Because we need a lot of indices in this paper, our use of $i$ to define $\omega$ is the only place in the paper where we will use $i$ to denote $\sqrt{-1}$.)

Let $F$ be the $n \times n$ matrix that represents the discrete Fourier transform: $F_{ij} = \omega^{ij}$. We use $I_n$ to denote the identity matrix of size $n \times n$. For $i \in [n]$, we use $e_i$ to represent the $i$th basis vector in $\mathbb{R}^n$. We use $\otimes$ to represent the Kronecker product between two matrices (which also refers to the outer product of two vectors).

## 2    ANALYSIS

### 2.1    ONE FILTER

As a warmup, we focus on the case that the number $m$ of input channels and output channels is 1. In this case, the filter coefficients are simply a $k \times k$ matrix. It will simplify notation, however, if we embed this $k \times k$ matrix in an $n \times n$ matrix, by padding with zeroes (which corresponds to the fact that the offsets with those indices are not used). Let us refer to this $n \times n$ matrix also as $K$.

---

[3] For two multisets $\mathcal{S}$ and $\mathcal{T}$, we use $\mathcal{S} \cup \mathcal{T}$ to denote the multiset obtained by summing the multiplicities of members of $\mathcal{S}$ and $\mathcal{T}$.

An $n^2 \times n^2$ matrix $A$ is *doubly block circulant* if $A$ is a circulant matrix of $n \times n$ blocks that are in turn circulant.

For a matrix $X$, let $\text{vec}(X)$ be the vector obtained by stacking the columns of $X$.

**Lemma 1 (see Jain (1989) Section 5.5, Goodfellow et al. (2016) page 329)** *For any filter coefficients $K$, the linear transform for the convolution by $K$ is represented by the following doubly block circulant matrix:*

$$A = \begin{bmatrix} \text{circ}(K_{0,:}) & \text{circ}(K_{1,:}) & \dots & \text{circ}(K_{n-1,:}) \\ \text{circ}(K_{n-1,:}) & \text{circ}(K_{0,:}) & \dots & \text{circ}(K_{n-2,:}) \\ \vdots & \vdots & \vdots & \vdots \\ \text{circ}(K_{1,:}) & \text{circ}(K_{2,:}) & \dots & \text{circ}(K_{0,:}) \end{bmatrix}. \tag{1}$$

*That is, if $X$ is an $n \times n$ matrix, and $Y$ is the result of a 2-d convolution of $X$ with $K$, i.e.*

$$\forall ij, \ Y_{ij} = \sum_{p \in [n]} \sum_{q \in [n]} X_{i+p,j+q} K_{p,q} \tag{2}$$

*then* $\text{vec}(Y) = A \, \text{vec}(X)$.

So now we want to determine the singular values of a doubly block circulant matrix.

We will make use of the characterization of the eigenvalues and eigenvectors of doubly block circulant matrices, which uses the following definition: $Q \stackrel{\text{def}}{=} \frac{1}{n}(F \otimes F)$.

**Theorem 2 (Jain (1989) Section 5.5)** *For any $n^2 \times n^2$ doubly block circulant matrix $A$, the eigenvectors of $A$ are the columns of $Q$.*

To get singular values in addition to eigenvalues, we need the following two lemmas.

**Lemma 3 (Jain (1989) Section 5.5)** $Q$ *is unitary.*

Using Theorem 2 and Lemma 3, we can get the eigenvalues as the diagonal elements of $Q^*AQ$.

**Lemma 4** *The matrix $A$ defined in equation 1 is normal, i.e., $A^T A = AA^T$.*

**Proof**:
$$AA^T = AA^* = Q^*DQQ^*D^*Q = Q^*DD^*Q = Q^*D^*DQ = Q^*D^*QQ^*DQ = A^*A = A^TA.$$
$\square$

The following theorem characterizes the singular values of $A$ as a simple function of $K$. As we will see, a characterization of the eigenvalues plays a major role. Chao (1974) provided a more technical characterization of the eigenvalues which may be regarded as making partial progress toward Theorem 5. However, we provide a proof from first principles, since it is the cleanest way we know to prove the theorem.

**Theorem 5** *For the matrix $A$ defined in equation 1, the eigenvalues of $A$ are the entries of $F^T K F$, and its singular values are their magnitudes. That is, the singular values of $A$ are*

$$\left\{ \left| (F^T K F)_{u,v} \right| \ : \ u,v \in [n] \right\}. \tag{3}$$

**Proof**: By Theorems 2 and Lemma 3, the eigenvalues of $A$ are the diagonal elements of $Q^*AQ = \frac{1}{n^2}(F^* \otimes F^*)A(F \otimes F)$. If we view $(F^* \otimes F^*)A(F \otimes F)$ as a compound $n \times n$ matrix of $n \times n$ blocks, for $u,v \in [n]$, the $(un+v)$th diagonal element is the $v$th element of the $u$th diagonal block. Let us first evaluate the $u$th diagonal block. Using $i,j$ to index blocks, we have

$$(Q^*AQ)_{uu} = \frac{1}{n^2} \sum_{i,j \in [n]} (F^* \otimes F^*)_{ui} A_{ij} (F \otimes F)_{ju} = \frac{1}{n^2} \sum_{i,j \in [n]} \omega^{-ui} F^* \text{circ}(K_{j-i,:}) \omega^{ju} F$$

$$= \frac{1}{n^2} \sum_{i,j \in [n]} \omega^{u(j-i)} F^* \text{circ}(K_{j-i,:}) F. \tag{4}$$

To get the $v$th element of the diagonal of (4), we may sum the $v$th elements of the diagonals of each of its terms. Toward this end, we have

$$(F^* \text{circ}(K_{j-i,:})F)_{vv} = \sum_{r,s \in [n]} \omega^{-vr} \text{circ}(K_{j-i,:})_{rs} \omega^{sv} = \sum_{r,s \in [n]} \omega^{v(s-r)} K_{j-i,s-r}.$$

Substituting into (4), we get $\frac{1}{n^2} \sum_{i,j,r,s \in [n]} \omega^{u(j-i)} \omega^{v(s-r)} K_{j-i,s-r}$. Collecting terms where $j - i = p$ and $s - r = q$, this is $\sum_{p,q \in [n]} \omega^{up} \omega^{vq} K_{p,q} = (F^T K F)_{uv}$.

Since the singular values of any normal matrix are the magnitudes of its eigenvalues (Horn and Johnson (2012) page 158), applying Lemma 4 completes the proof. $\square$

Note that $F^T K F$ is the 2D Fourier transform of $K$, and recall that $||A||_2$ is the largest singular value of $A$.

## 2.2 MULTI-CHANNEL CONVOLUTION

Now, we consider case where the number $m$ of channels may be more than one. Assume we have a 4D kernel tensor $K$ with element $K_{p,q,c,d}$ giving the connection strength between a unit in channel $d$ of the input and a unit in channel $c$ of the output, with an offset of $p$ rows and $q$ columns between the input unit and the output unit. The input $X \in \mathbb{R}^{m \times n \times n}$; element $X_{d,i,j}$ is the value of the input unit within channel $d$ at row $i$ and column $j$. The output $Y \in \mathbb{R}^{m \times n \times n}$ has the same format as $X$, and is produced by

$$Y_{crs} = \sum_{d \in [m]} \sum_{p \in [n]} \sum_{q \in [n]} X_{d,r+p,s+q} K_{p,q,c,d}. \tag{5}$$

By inspection, $\text{vec}(Y) = M \text{vec}(X)$, where $M$ is as follows

$$M = \begin{bmatrix} B_{00} & B_{01} & \dots & B_{0(m-1)} \\ B_{10} & B_{11} & \dots & B_{1(m-1)} \\ \vdots & \vdots & \dots & \vdots \\ B_{(m-1)0} & B_{(m-1)1} & \dots & B_{(m-1)(m-1)} \end{bmatrix} \tag{6}$$

and each $B_{cd}$ is a doubly block circulant matrix from Lemma 1 corresponding to the portion $K_{:,:,c,d}$ of $K$ that concerns the effect of the $d$-th input channel on the $c$-th output channel. (We can think of each output in the multichannel case as being a sum of single channel filters parameterized by one of the $K_{:,:,c,d}$'s.)

The following is our main result.

**Theorem 6** *For any $K \in \mathbb{R}^{n \times n \times m \times m}$, let $M$ is the matrix encoding the linear transformation computed by a convolutional layer parameterized by $K$, defined as in (6). For each $u, v \in [n] \times [n]$, let $P^{(u,v)}$ be the $m \times m$ matrix given by $P_{cd}^{(u,v)} = (F^T K_{:,:,c,d} F)_{uv}$. Then*

$$\sigma(M) = \bigcup_{u \in [n], v \in [n]} \sigma\left(P^{(u,v)}\right). \tag{7}$$

The rest of this section is devoted to proving Theorem 6 through a series of lemmas.

The analysis of Section 2.1 implies that for all $c, d \in [m]$, $D_{cd} \overset{\text{def}}{=} Q^* B_{cd} Q$ is diagonal. Define

$$L \overset{\text{def}}{=} \begin{bmatrix} D_{00} & D_{01} & \dots & D_{0(m-1)} \\ D_{10} & D_{11} & \dots & D_{1(m-1)} \\ \vdots & \vdots & \dots & \vdots \\ D_{(m-1)0} & D_{(m-1)1} & \dots & D_{(m-1)(m-1)} \end{bmatrix}. \tag{8}$$

**Lemma 7** *$M$ and $L$ have the same singular values.*

**Proof**: We have

$$
M = \begin{bmatrix} B_{00} & \cdots & B_{0(m-1)} \\ \vdots & \vdots & \vdots \\ B_{(m-1)0} & \cdots & B_{(m-1)(m-1)} \end{bmatrix} = \begin{bmatrix} QD_{00}Q^* & \cdots & QD_{0(m-1)}Q^* \\ \vdots & \vdots & \vdots \\ QD_{(m-1)0}Q^* & \cdots & QD_{(m-1)(m-1)}Q^* \end{bmatrix}
$$

$$
= R \begin{bmatrix} D_{00} & \cdots & D_{0(m-1)} \\ \vdots & \vdots & \vdots \\ D_{(m-1)0} & \cdots & D_{(m-1)(m-1)} \end{bmatrix} R^* = RLR^*,
$$

where $R \stackrel{\text{def}}{=} I_m \otimes Q$. Note that $R$ is unitary because

$$
RR^* = (I_m \otimes Q)(I_m \otimes Q^*) = (I_m I_m) \otimes (QQ^*) = I_{mn^2};
$$

this implies that $M$ and $L$ have the same singular values. □

So now we have as a subproblem characterizing the singular values of a block matrix whose blocks are diagonal. To express the the characterization, it helps to reshape the nonzero elements of $L$ into a $m \times m \times n^2$ tensor $G$ as follows: $G_{cdw} = (D_{cd})_{ww}$.

**Theorem 8** $\sigma(L) = \bigcup\limits_{w \in [n^2]} \sigma\left(G_{:,:,w}\right).$

**Proof**: Choose an arbitrary $w \in [n^2]$, and a (scalar) singular value $\sigma$ of $G_{:,:,w}$ whose left singular vector is $x$ and whose right singular vector is $y$, so that $G_{:,:,w}y = \sigma x$. Recall that $e_w \in \mathbb{R}^{n^2}$ is a standard basis vector.

We claim that $L(y \otimes e_w) = \sigma(x \otimes e_w)$. Since $D_{cd}$ is diagonal, $D_{cd}e_w = (D_{cd})_{ww}e_w = G_{cdw}e_w$. Thus we have $(L(y \otimes e_w))_c = \sum_{d \in [m]} D_{cd}y_d e_w = (\sum_{d \in [m]} G_{cdw}y_d)e_w = (G_{:,:,w}y)_c e_w = \sigma x_c e_w$, which shows that

$$
L(y \otimes e_w) = \begin{bmatrix} D_{00} & \cdots & D_{0(m-1)} \\ D_{10} & \cdots & D_{1(m-1)} \\ \vdots & \cdots & \vdots \\ D_{(m-1)0} & \cdots & D_{(m-1)(m-1)} \end{bmatrix} \begin{bmatrix} y_0 e_w \\ \vdots \\ y_{m-1} e_w \end{bmatrix} = \begin{bmatrix} \sigma x_0 e_w \\ \vdots \\ \sigma x_{m-1} e_w \end{bmatrix} = \sigma(x \otimes e_w).
$$

If $\tilde{\sigma}$ is another singular value of $G_{:,:,w}$ with a left singular vector $\tilde{x}$ and a right singular vector $\tilde{y}$, then $\langle (x \otimes e_w), (\tilde{x} \otimes e_w) \rangle = \langle x, \tilde{x} \rangle = 0$ and, similarly $\langle (y \otimes e_w), (\tilde{y} \otimes e_w) \rangle = 0$. Also, $\langle (x \otimes e_w), (x \otimes e_w) \rangle = 1$ and $\langle (y \otimes e_w), (y \otimes e_w) \rangle = 1$.

For any $x$ and $\tilde{x}$, whether they are equal or not, if $w \neq \tilde{w}$, then $\langle (x \otimes e_w), (\tilde{x} \otimes e_{\tilde{w}}) \rangle = 0$, simply because their non-zero components do not overlap.

Thus, by taking the Kronecker product of each singular vector of $G_{:,:,w}$ with $e_w$ and assembling the results for various $w$, we may form a singular value decomposition of $L$ whose singular values are $\cup_{w \in [n^2]} \sigma(G_{:,:,w})$. This completes the proof. □

Using Lemmas 7 and Theorem 8, we are now ready to prove Theorem 6.

**Proof (of Theorem 6)**. Recall that, for each input channel $c$ and output channel $d$, the diagonal elements of $D_{c,d}$ are the eigenvalues of $B_{c,d}$. By Theorem 5, this means that the diagonal elements of $D_{c,d}$ are

$$
\{(F^T K_{:,:,c,d} F)_{u,v} : u, v \in [n]\}. \tag{9}
$$

The elements of (9) map to the diagonal elements of $D_{c,d}$ as follows:

$$
G_{cdw} = (D_{c,d})_{ww} = (F^T K_{:,:,c,d} F)_{\lfloor w/m \rfloor, w \mod m}
$$

and thus

$$
G_{:,:,w} = \left((F^T K_{:,:,c,d} F)_{\lfloor w/m \rfloor, w \mod m}\right)_{cd}
$$

which in turn implies

$$
\bigcup\limits_{w \in [n^2]} \sigma(G_{:,:,w}) = \cup_{u \in [n], v \in [n]} \sigma\left(\left((F^T K_{:,:,c,d} F)_{u,v}\right)_{cd}\right).
$$

Applying Lemmas 7 and 8 completes the proof. □

## 3 REGULARIZATION

We now show how to use the spectrum computed above to project a convolution onto the set of convolutions with bounded operator norm. We exploit the following key fact.

**Proposition 9 (Lefkimmiatis et al. (2013), Proposition 1)** *Let $A \in \mathbb{R}^{n \times n}$, and let $A = UDV^\top$ be its singular value decomposition. Let $\tilde{A} = U\tilde{D}V^\top$, where, for all $i \in [n]$, $\tilde{D}_{ii} = \min(D_{ii}, c)$ and $\mathcal{B} = \{X \mid ||X||_2 \leq c\}$. Then $\tilde{A}$ is the projection of $A$ onto $\mathcal{B}$; i.e. $\tilde{A} = \underset{X \in \mathcal{B}}{\arg\min} ||A - X||_F$.*

This implies that the desired projection can be obtained by clipping the singular values of linear transformation associated with a convolutional layer to the interval $[0, c]$. Note that the eigenvectors remained the same in the proposition, hence the projected matrix is still generated by a convolution. However, after the projection, the resulting convolution neighborhood may become as large as $n \times n$. On the other hand, we can project this convolution onto the set of convolutions with $k \times k$ neighborhoods, by zeroing out all other coefficients. NumPy code for this is in Appendix A.

Repeatedly alternating the two projections would give a point in the intersection of the two sets, i.e., a $k \times k$ convolution with bounded operator norm (Cheney and Goldstein (1959) Theorem 4, Boyd and Dattorro (2003) Section 2), and the projection onto that intersection could be found using the more complicated Dykstra's projection algorithm (Boyle and Dykstra, 1986).

When we wish to control the operator norm during an iterative optimization process, however, repeating the alternating projections does not seem to be worth it – we found that the first two projections already often produced a convolutional layer with an operator norm close to the desired value. Furthermore, because SGD does not change the parameters very fast, we can think of a given pair of projections as providing a warm start for the next pair.

In practice, we run the two projections once every few steps, thus letting the projection alternate with the training.

## 4 EXPERIMENTS

First, we validated Theorem 6 with unit tests in which the output of the code given in the introduction is compared with evaluating the singular values by constructing the full matrix encoding the linear transformation corresponding to the convolutional layer and computing its SVD.

### 4.1 TIMING

We generated 4D tensors of various shapes with random standard normal values, and computed their singular values using the full matrix method, the NumPy code given above and the equivalent TensorFlow code. For small tensors, the NumPy code was faster than TensorFlow, but for larger tensors, the TensorFlow code was able to exploit the parallelism in the algorithm and run much faster on a GPU. The timing results are shown in Figure 1.

### 4.2 REGULARIZATION

We next explored the effect of regularizing the convolutional layers by clipping their operator norms as described in Section 3. We ran the CIFAR-10 benchmark with a standard 32 layer residual network with 2.4M training parameters; (He et al., 2016). This network reached a test error rate of $6.2\%$ after 250 epochs, using a learning rate schedule determined by a grid search (shown by the gray plot in Figure 2). We then evaluated an algorithm that, every 100 steps, clipped the norms of the convolutional layers to various different values between 0.1 and 3.0. As expected, clipping to 2.5 and 3.0 had little impact on the performance, since the norms of the convolutional layers were between 2.5 and 2.8. Clipping to 0.1 yielded a surprising $6.7\%$ test error, whereas clipping to 0.5 and 1.0 yielded test errors of $5.3\%$ and $5.5\%$ respectively (shown in Figure 2). A plot of test error against training time is provided in Figure 4 in Appendix B, showing that the projections did not slow down the training very much.

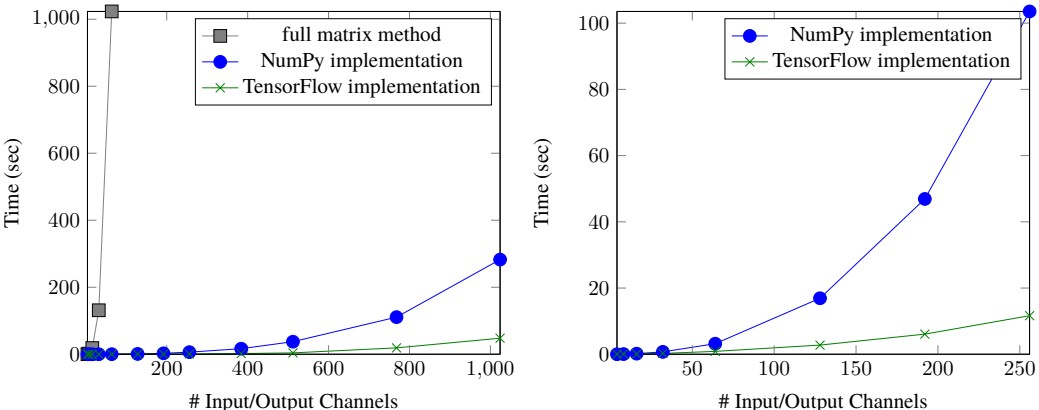

Figure 1: Time used to compute singular values. The left graph is for a $3 \times 3$ convolution on a $16 \times 16$ image with the number of input/output channels on the $x$-axis. The right graph is for a $11 \times 11$ convolution on a $64 \times 64$ image (no curve for full matrix method is shown as this method could not complete in a reasonable time for these inputs).

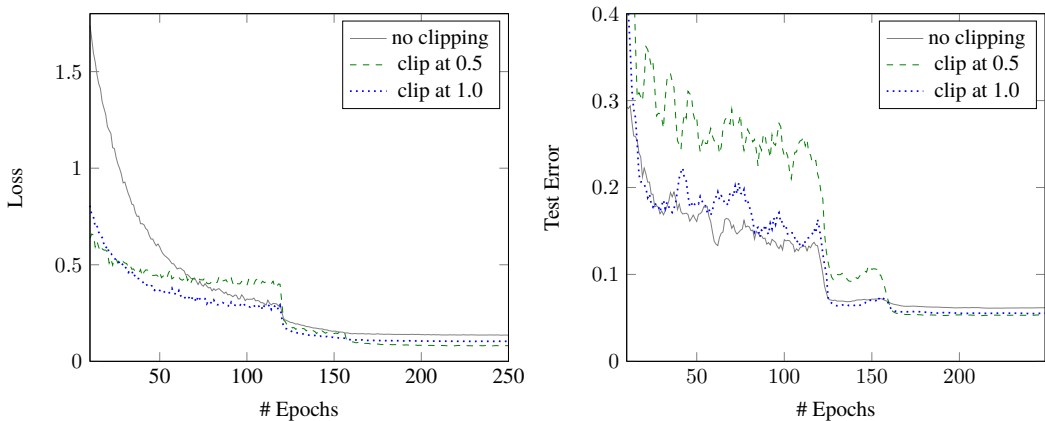

Figure 2: Training loss and test error for ResNet model (He et al., 2016) for CIFAR-10.

### 4.3 ROBUSTNESS TO CHANGES IN HYPERPARAMETERS

The baseline algorithm studied in the previous subsection used batch normalization. Batch normalization tends to make the network less sensitive to linear transformations with large operator norms. However, batch normalization includes trainable scaling parameters (called $\gamma$ in the original paper) that are applied after the normalization step. The existence of these parameters lead to a complicated interaction between batch normalization and methods like ours that act to control the norm of the linear transformation applied before batch normalization.

Because the effect of regularizing the operator norm is more easily understood in the absence of batch normalization, we also performed experiments with a baseline that did not use batch normalization.

Another possibility that we wanted to study was that using a regularizer may make the process overall more stable, enabling a larger learning rate. We were generally interested in whether operator-norm regularization made the training process more robust to the choice of hyperparameters.

In one experiment, we started with the same baseline as the previous subsection, but disabled batch normalization. This baseline started with a learning rate of 0.1, which was multiplied by a factor 0.95 after every epoch. We tried all combinations of the following hyperparameters: (a) the norm of the ball projected onto (no projection, 0.5, 1.0, 1.5, 2.0); (b) the initial learning rate (0.001, 0.003, 0.01, 0.03, 0.1); (c) the minibatch size (32, 64); (d) the number of epochs per decay of the learning

rate (1,2,3). We tried each of the 150 combinations of the hyperparameters, trained for 100 epochs, and measured the test error. The results are plotted in Figure 3a. The operator norm regularization

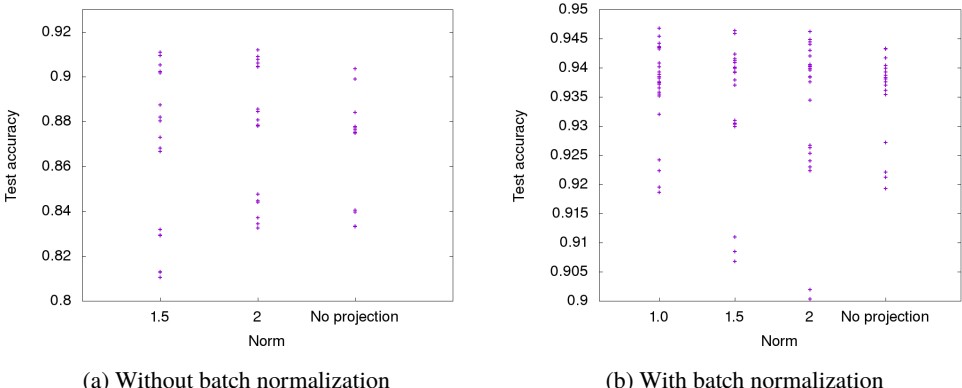

(a) Without batch normalization

(b) With batch normalization

Figure 3: A scatterplot of the test errors obtained with different hyperparameter combinations, and different operator-norm regularizers.

improved the best result, and also made the process more robust to the choice of hyperparameters.

We conducted a similar experiment in the presence of batch normalization, except using learning rates 0.01, 0.03, 0.1, 0.2, and 0.3. Those results are shown in Figure 3b. Regularizing the operator norm helps, even in the presence of batch normalization.

It appears that operator-norm regularization and batch normalization are not redundant, and neither dominates the other. We were surprised by this.

## 4.4 Comparison with reshaping $K$

In Section 1 we mentioned that Yoshida and Miyato (2017) approximated the linear transformation induced by $K$ by reshaping $K$. This leads to an alternate regularization method — we compute the spectrum of the reshaped $K$, and project it onto a ball using clipping, as above. We implemented this an experimented with it using the same network and hyperparameters as in Section 4.2 and found the following.

- We clipped the singular values of the reshaped $K$ every 100 steps. We tried various constants for the clipped value (0.05, 0.1, 0.2, 0.5, 1.0), and found that the best accuracy we achieved, using 0.2, was the same as the accuracy we achieved in Section 4.2.
- We clipped the singular values of the reshaped $K$ to these same values every step, and found that the best accuracy achieved was slightly worse than the accuracy achieved in the previous step. We observed similar behavior when we clipped norms using our method.
- Most surprisingly, we found that clipping norms by our method on a GPU was about 25% faster than clipping the singular values of the reshaped $K$ — when we clipped after every step, on the same machine, 10000 batches of CIFAR10 took 14490 seconds when we clipped the reshaped $K$, whereas they took 11004 seconds with our exact method! One possible explanation is parallelization — clipping reshaped $K$ takes $O(m^3 k^2)$ flops, whereas our method does $m^2$ FFTs, followed by $n^2$ $m \times m$ SVDs, which takes $O(m^3 n^2)$ flops, but these can be parallelized and completed in as little as $O(n^2 \log n + m^3)$ time.

Clearly this is only one dataset, and the results may not generalize to other sets. However it does suggest that finding the full spectrum of the convolutional layer may be no worse than computing heuristic approximations, both in classification accuracy and speed.

## 5 Acknowledgements

We thank Tomer Koren, Nishal Shah, Yoram Singer and Chiyuan Zhang for valuable conversations.

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

## A NUMPY CODE FOR OPERATOR NORM PROJECTION

```
def Clip_OperatorNorm(kernel, input_shape, clip_to):
  transform_coefficients = np.fft.fft2(kernel, input_shape, axes=[0, 1])
  U, D, V = np.linalg.svd(transform_coefficients, compute_uv=True, full_matrices=False)
  D_clipped = np.minimum(D, clip_to)
  if kernel.shape[2] > kernel.shape[3]:
    clipped_transform_coefficients = np.matmul(U, D_clipped[..., None] * V)
  else:
    clipped_transform_coefficients = np.matmul(U * D_clipped[..., None, :], V)
  clipped_kernel = np.fft.ifft2(clipped_transform_coefficients, axes=[0, 1]).real
  return clipped_kernel[np.ix_(*[range(d) for d in kernel.shape])]
```

## B TEST ERROR VS. TRAINING TIME

Figure 4 shows the plots of test error vs. training time in our CIFAR-10 experiment.

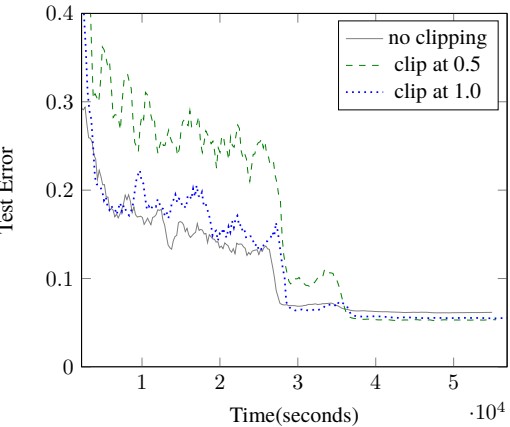

Figure 4: Test error vs. training time for ResNet model (He et al., 2016) for CIFAR-10.

## C THE OFFICIAL PRE-TRAINED RESNET MODEL

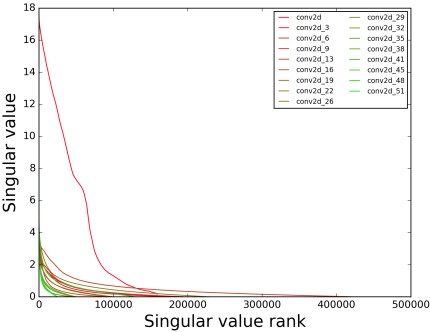

Figure 5: Plot of the singular values of the linear operators associated with the convolutional layers of the pretrained "ResNet V2" from the TensorFlow website.

The singular values of the convolutional layers from the official "Resnet V2" pre-trained model (He et al., 2016) are plotted in Figure 5. The singular values are ordered by value. Only layers with kernels larger than $1 \times 1$ are plotted. The curves are plotted with a mixture of red and green; layers

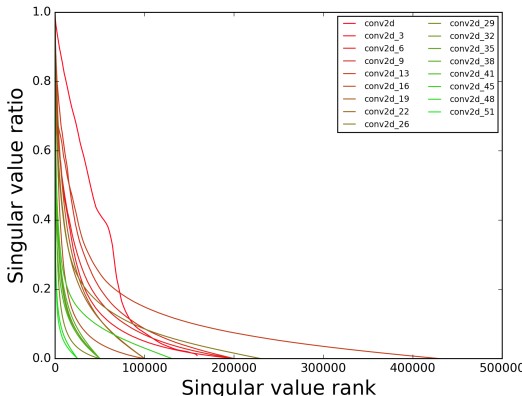

Figure 6: Plot of the ratio of singular values to maximum singular value of the linear operators associated with the convolutional layers of the pretrained "ResNet V2" from the TensorFlow website.

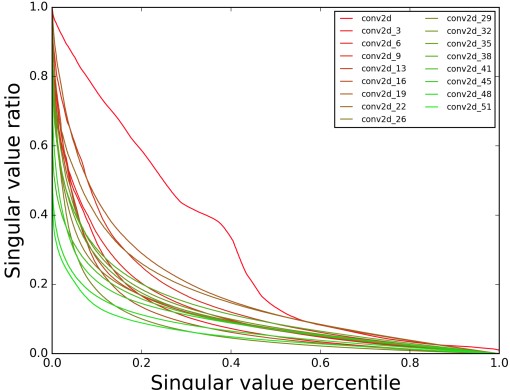

Figure 7: Plot of the ratio of singular values to maximum singular value of the linear operators associated with the convolutional layers of the pretrained "ResNet V2" normalized by size of the convolution.

closer to the input are plotted with colors with a greater share of red. The transformations with the largest operator norms are closest to the inputs. As the data has undergone more rounds of processing, as we proceed through the layers, the number of non-negligible singular values increases for a while, but at the end, it tapers off.

In Figure 5, we plotted the singular values ordered by value. It can be observed that while singular values in the first layer are much larger than the rest, many layers have a lot of singular values that are pretty big. For example, most of the layers have at least 10000 singular values that are at least 1. To give a complementary view, Figure 6 presents a plot of the ratios of the singular values in each layer with the largest singular value in that layer. We see that the effective rank of the convolutional layers is larger closer to the inputs.

Figure 6 shows that different convolutional layers have significantly different numbers of non-negligible singular values. A question that may arise is to what extent this was due to the fact that different layers simply are of different sizes, so that the total number of their singular values, tiny or not, was different. To look into this, instead of plotting the singular value ratios as a function of the rank of the singular values, as in the Figure 6, we normalized the values on the horizontal axis by dividing by the total number of singular values. The result is shown in Figure 7.

