# OpenReview forum: "The Singular Values of Convolutional Layers"
_ICLR.cc/2019/Conference_

### Official Review · AnonReviewer3 · 2018-10-25

**Rating:** 7
**Confidence:** 3

**Review:**

In this paper, the authors derive exact formulas for computing singular values of convolutional layers of deep neural networks. By appealing to fast FFT transformations, they show that computing the singular values can be done much faster than computing the full SVD of the convolution matrix. This obviates the needs to approximate the singular values. They use these results to then devise regularization schemes for DNN layers, and show that employing this regularization helps with model performance.

They show that the algorithm with the operator norm regularization can be solved via an alternating projection scheme. They also postulate that since this might be expensive and unnecessary, one can also perform just 2 projections after every few SG iterations, and claim that this acts as a 'warm start' for subsequent iterations. Experiments reveal that this does not degrade the performance too much.


The paper is well written and easy to understand. The proofs follow from standard linear algebra methods, and are easy to follow.

---

> ### Author Response · Authors · 2018-11-06
> **thank you**
>
> Thank you for your kind review.

---

### Official Review · AnonReviewer2 · 2018-10-29

**Rating:** 4
**Confidence:** 5

**Review:**

This paper studies the problem on computing the spectrum of singular values of linear convolutional layers. This is an important problem with abundant applications on regularizing deep neural networks. However, there are several technical issues need to be addressed in its current form.

First, in the section "Summary of Results", at first read of the paper I found it very confusing why the time complexity of computing the spectrum is a function of n, where n is the size of the input feature map. Intuitively, since the size of the convolutional kernel is m x m x k x k, it is expected that the time complexity is expressed as a function of (m, k). Later I realized that this is due to the unnecessary and redundant 0 padding in section 2.1 that leads to this artifact. I understand that in order to apply the described Fourier transform technique it is necessary to introduce the large nxn filter, which is of the same size as the input, but it also introduces redundant computation. This fact further emerges in the introduction of matrix A in Eq (1).

More importantly, I think the authors didn't perform a detailed analysis on using the basic definition of convolutional filter to compute its spectrum, and this is the reason why they reached a misleading conclusion that simple SVD takes O(n^6 m^3) time. Specifically, each convolution operation corresponds to a inner product operation, so we can reshape the input 3D tensor with shape m x n x n into a 2D matrix, with shape n^2 x mk^2, denoted as X. Note that this creates a unnecessary redundancy in the input feature map, but it does not create redundant weight for the convolutional kernel. As a comparison, the introduced matrix A in the paper is heavily redundant. Similarly, for m channels, we can reshape the 4D convolutional kernel with shape m x m x k x k into a 2D matrix, with shape mk^2 x m, denoted as K. Then the usual convolution layer can be described as the following linear system: Y = X K, where Y with shape n^2 x m is the output, and can be easily reshaped into size m x n x n. Hence to compute the spectrum of the convolution layer corresponds to computing the singular values of the 2D matrix K with size mk^2 x m. Hence a naive application of SVD directly gives us the solution in time O(m^3 k^2) (Note that the time complexity of SVD for matrix with size a x b is O(min{a^2 b, a b^2})), which is much smaller than the one given in the paper O(m^3 n^2) since k << n.

In experiment the authors made unfair comparison between their proposed method and the full matrix method: the full matrix A is fully redundant, due to its circulant pattern. As this implies a highly redundant information, nobody will form and compute matrix A explicitly in practice. So the time improvements demonstrated in the experiment section are meaningless. A valid baseline would be to compare the proposed method with the one introduced above. But in this case I would imagine the proposed method to be worse due to its unnecessary 0 padding leading to the worst time complexity.

---

> ### Author Response · Authors · 2018-11-06
> **the singular values of the reshaped kernel cannot be correct**
>
> It can be proved that if a kernel of size (k, k, m, m) is applied to an input of size (n, n, m), and the entries of the kernel are chosen from a continuous probability distribution like a Gaussian, then with probability 1, the number of distinct singular values in the linear transformation of its convolutional layer is at least (m n^2)/2.   Repeatedly applying the code on page two of our paper to random inputs has always produced at least this number of distinct singular values.  (We invite the reader to try this.)  Any reshaping of an (k, k, m, m) kernel can produce at most mk singular values, which is not enough to be correct.  The reshaping outlined in your review can have at most m singular values, which is even smaller.
>
> As we noted in our answer to a question during the review period, an example in which the reshaping does not compute even the largest singular value correctly can be obtained by executing the following commands after importing numpy as np and defining SingularValues as in page two of our paper:
>
>    kernel = np.array(np.ones(4)).reshape(2,2,1,1)
>    SingularValues(kernel, [4,4])
>    reshaped = kernel.reshape(4,1)
>    np.linalg.svd(reshaped, compute_uv=False)
>
> The correct largest singular value of the convolutional layer is 4. (An all-ones feature map is turned into an all-fours feature map by applying this filter.) The (only) singular value of the reshaping of the kernel, which is (1,1,1,1)^T, is 2.  This is not an isolated instance --- random inputs also produce counterexamples --- we ran our algorithm with m = 1, n = 16, and various values of k. In each case we found 130 unique singular values, with ranges described below. The reshaping method produced 1 singular value in each case:
> k                Range of Singular Values                     Singular Value from reshaping
> 4                [0.388872696514, 7.6799308322]        3.76770352
> 5                [0.421739253704, 10.7721306924]      5.04019121
> 7                [0.165159699404, 14.9556902191]      6.7304902
> 7                [0.532454839614, 16.38084538]          7.20513578
> 3                [0.4655241798,  6.12218595024]         3.48952531
> 6                [0.418950277234, 15.2821360286]      6.1289447

---

> > ### Comment · AnonReviewer2 · 2018-11-06
> > **Unclear definition of linear map**
> >
> > Thanks for the response. It seems that the confusion comes from the definition of the linear map and its singular values induced by the convolutional kernel. Let me ask the problem in the following way: what is the actual linear function that being applied to the input tensor in CNNs? Given a 4D convolutional tensor with shape (k, k, m, m), for each fixed slice in the 4th dimension, the linear map is defined by the 3D tensor from the first 3 dimensions, as it is this 3D tensor that is being taken inner product repeatedly with each part of the input 3D tensor. Hence the reshape given in my first comment exactly reflect this fact.
> >
> > Put it in another way, the inner product between two matrices of same dimension A, B is defined as Tr(A^TB), but this defines the same Euclidean geometry by saying that Tr(A^T B) = <vec(A), vec(B)>. In the case of CNN we are dealing with 4D tensors, but the actual linear map is of shape mk^2 x m, as the first three dimensions of each slice in the 4th dimension are used to define the linear map.
> >
> > The argument of sampling kernel weights from a continuous distribution is irrelevant. As a bottomline, the complexity of computing the spectrum of the linear map (the convolutional kernel) itself shouldn't have anything to do with the input size n.

---

> > > ### Author Response · Authors · 2018-11-07
> > > **analyzing the convolutional layer as a whole**
> > >
> > > We agree that the reviewer has correctly described the operational implementation of a convolutional layer --- a set of k x k patches is created from the input, and the reshaped convolution matrix is applied to each patch to get the output.  However, the subject of our paper is the computation of the singular values of the (linear) function computed by the convolutional layer as a whole, which determine its potential contribution to exploding and vanishing gradients.  This is a linear map applied to the n^2 m inputs that produces n^2 m outputs, and we compute the singular values of the n^2 m x n^2 m matrix corresponding to this linear map.

---

> > > > ### Comment · AnonReviewer2 · 2018-11-08
> > > > **Not the actual linear map in CNN, essentially reduce to fully-connected layer**
> > > >
> > > > I appreciate the authors' response, as it makes the whole procedure proposed in the paper much more clear. If my understanding is correct, then basically what being done here is to vectorize both the input and output 3D tensors and then treat them as two fully-connected layers. So the linear map computed in the paper corresponds to the fully connected matrix here, which also explains why there is an input size n in the time complexity of the proposed method.
> > > >
> > > > But this is not the actual linear map happened in CNNs, as the same convolutional kernel is being applied repeatedly, i.e., the weights are tied. Perhaps what's more problematic is that, although the huge matrix A has the circulant pattern (actually this is not true in practice as well, because the circulant pattern is due to the assumption of wrapping around), when computing the singular values of A, different blocks are treated as independent, but in fact those weights are being tied. In other words, when computing the gradient of this kernel, using this method (by vectorzing both 3D tensors into vectors and compute the gradient of the huge matrix A) will lead to the wrong gradient. The correct one is to re-add different blocks of the gradient w.r.t. A according to the circulant pattern.
> > > >
> > > > Intuitively, what I am saying is that in CNNs the linear map is unique, and the redundancy comes from the overlapping patches in the input 3D tensor, while what the proposed method computes is the other way around.
> > > >
> > > > I implemented the proposed method and compared it with the one I gave above, and the results confirm my analysis:
> > > >
> > > > def SingularValues(kernel, input_shape):
> > > >     transform_coefficients = np.fft.fft2(kernel, input_shape, axes=[0, 1])
> > > >     return np.linalg.svd(transform_coefficients, compute_uv=False)
> > > >
> > > >
> > > > def Reshaped(kernel):
> > > >     matrix = np.reshape(kernel, (-1, kernel.shape[-1]))
> > > >     return np.linalg.svd(matrix, compute_uv=False)
> > > >
> > > >
> > > > def check(m, k):
> > > >     """
> > > >     m = # chanels
> > > >     k = # kernel sizes
> > > >     """
> > > >     # input shape
> > > >     n = 28
> > > >     kernel = np.random.rand(k, k, m, m)
> > > >     start_time = time.time()
> > > >     svs = SingularValues(kernel, (n, n))
> > > >     end_time = time.time()
> > > >     print("Time used by the proposed method: {} seconds".format(end_time - start_time))
> > > >     start_time = time.time()
> > > >     mines = Reshaped(kernel)
> > > >     end_time = time.time()
> > > >     print("Time used by the reshaping method: {} seconds".format(end_time - start_time))
> > > >     print("Shape = {}, Singular Values returned by the proposed method: {}".format(svs.shape, svs))
> > > >     print("Shape = {}, Singular Values computed by reshaping: {}".format(mines.shape, mines))
> > > >
> > > >
> > > > check(64, 5)
> > > > ---------------------------------------------------------------------------------------------
> > > > The singular values computed by these two different methods are different, and the proposed method takes around 0.4 second to finish while the reshaping one takes 0.02.

---

> > > > > ### Author Response · Authors · 2018-11-08
> > > > > **not fully connected**
> > > > >
> > > > > We thank the reviewer for the effort and time invested in this review. However it is clear that still quite a few misunderstandings of our work remain.  A convolutional layer applied to an n x n feature map with m input channels and m output channels is a function with n^2 m inputs and n^2 m outputs.  This function is linear.  The standard encoding of a linear function as a matrix has n^2 m rows and n^2 m columns.  The singular values of a linear function are defined to be the singular values of this matrix.  Computing these singular values is the subject of this paper.  The matrix and its singular values are a property of the function computed by the layer, no matter how it is implemented. Due to the special nature of the linear transform induced by the convolution layer, the matrix does indeed have lots of redundancy; we exploit this in computing the singular values efficiently (see Equations (1) and (6)).  We have proved that our method provides the correct singular values, and further verified this with unit tests.  We have also proved, in our first response to your review, that no reshaping of the kernel tensor can possibly provide the correct singular values.  We are therefore not surprised that our method gives different answers than a reshaping of the kernel tensor.

---

> > > > > > ### Comment · AnonReviewer2 · 2018-11-10
> > > > > > **Different from the true linear map**
> > > > > >
> > > > > > Thanks for the response. I understand that the linear function studied in this paper is a linear map from n^2m to n^2m, characterized by the matrix A in the paper, and I also understand that the paper proposes to compute the singular values of this matrix A. This is interesting byitself. However, as I described in my last several threads, it is misleading and confusing to call the singular values of this matrix A "the singular values of the convolution layer". The actual linear map computed by a convolution layer is the one obtained by reshaping it to a 2D matrix, and it is the singular values of this matrix that have something to do with gradient vanishing or explosion because the gradient computed by the backpropagation is w.r.t. this 2D matrix (reshaped from the original 4D tensor), NOT the one (the matrix A in this paper) studied in this paper. It is not clear to me why people may be interested in computing the singular values of this redundant matrix A. Furthermore, the complexity of computing the singular values of matrix A is much worse (O(m^3n^2) >> O(m^3k^2)) than that of computing the true singular values of the actual linear map. Given the above two reasons, I will keep my current rating.

---

> > > > > > > ### Public Comment · (anonymous) · 2018-11-14
> > > > > > > **The simple circular 1D convolution**
> > > > > > >
> > > > > > > As far as I can see, reviewer's 2 main concern is that the singular values of relevance are the ones computed through a direct SVD from the underlying reshape and not the one where structure has been introduced to underlying representation of the linear map.
> > > > > > >
> > > > > > > I would like to ask reviewer 2 about this particular simple example to see his/her opinion.
> > > > > > >
> > > > > > > Consider a 1D filter x to be convolved with a signal y, both of which are of size n. The convolution performed here is a standard circular convolution. Such an operation can be represented as:
> > > > > > >
> > > > > > > z = X * y
> > > > > > >
> > > > > > > where X is a circulant matrix of the filter x. The authors of this paper are directly studying the singular values of this well structured matrix of the filter x. Of course in the setting where X is the underlying matrix that is equivalent to performing a convolutional layer but that is irrelevant in the context of my question.
> > > > > > >
> > > > > > > On the other hand, the same operation can be performed as
> > > > > > >
> > > > > > > z = Y *x
> > > > > > >
> > > > > > > where Y, the input signal, has the circulant structure instead of the filters.
> > > > > > >
> > > > > > > Reviewer 2 is arguing that the singular values of interests for vanishing gradients etc, are the singular values of the vector x. That is the singular values of the filter coming from the reshape without introducing any structure. According to this, there is a single singular value since x is a vector.
> > > > > > >
> > > > > > > However, in the authors perspective the singular values of interest are the ones from the first example. That is the singular values of the structured matrix X which in this case will yield n singular values as opposed to a single singular value.
> > > > > > >
> > > > > > > Is this the fundamental argument of reviewer 2?
> > > > > > >
> > > > > > > If yes, then this brings a contradiction. This is because if we handle the argument from operator perspective. For a linear operators T:V > V on an n dimensional finite vector space, x is an eigenvector of T if there exsits a \lambda such that T(x) = \lambda * x.
> > > > > > >
> > > > > > > Now, consider the case where the linear operator, T, can have the basis representation of a filter of all ones. Applying this on a signal x of all ones we get, T(x) = n * x. That means, we know that for such a filter n has to be an eigenvalue of the operator T.
> > > > > > >
> > > > > > > Now, following the authors approach, that is z = T(x) = F * x. The eigen values of F which is the matrix representation of T under appropriate basis will be indeed n. This is since F is a circulant matrix of all ones and the eigenvalues are a single n and (n-1) zero eigenvalues. However, according the reviewer, the eigen values of interest are the eigen values of f where T(x) = X * f.  However, the SVD of f (where f is a vector of ones) does not have a singular value of n in general. You can take several counter examples.
> > > > > > >
> > > > > > > Thus, according the definition of eigen decomposition on linear operators on finite dimensional space, the singular values of the structured matrix are the correct singular values of interest. More precisely, the singular values computed in this paper are indeed the singular values of the linear operator described as CNN convolution. I would love to hear back R2's comments and thoughts.
> > > > > > >
> > > > > > > Moreover, may the authors comment more clearly and cite the reference of this claim?
> > > > > > >
> > > > > > > "It can be proved that if a kernel of size (k, k, m, m) is applied to an input of size (n, n, m), and the entries of the kernel are chosen from a continuous probability distribution like a Gaussian, then with probability 1, the number of distinct singular values in the linear transformation of its convolutional layer is at least (m n^2)/2."
> > > > > > >
> > > > > > > As determining the correct number of singular values for a generic linear map (without matrix representation), i.e. without introducing a basis, will support their claims further.

---

> > > > > > > > ### Author Response · Authors · 2018-11-16
> > > > > > > > **uploaded proof of n^2/2 lower bound**
> > > > > > > >
> > > > > > > > We have uploaded a proof that, in the case of a single input channel and a single output channel, for random 2 x 2 filter applied to an n x n signal, with probability 1, the number of singular values in the resulting linear transformation is at least n^2/2.  This proof may be found in the document titled pairs2d.pdf in https://www.dropbox.com/sh/l8adgttixljdpz5/AADV_n6uxBFSX2q_0B7J2sIza
> > > > > > > >
> > > > > > > > Running the following code after importing numpy as np and defining SingularValues as in page two of our paper may also convince you that there are almost always at least n^2/2 singular values.
> > > > > > > >
> > > > > > > > for _ in range(20)
> > > > > > > >    	kernel = np.random.randn(2,2,1,1)
> > > > > > > >          input_shape = [4,4]
> > > > > > > >          print np.sort(SingularValues(kernel, input_shape).flatten())

---

> > > > > > > > > ### Public Comment · (anonymous) · 2018-11-20
> > > > > > > > > **Link does not work**
> > > > > > > > >
> > > > > > > > > May the authors update the link? It does not work.

---

> > > > > > > > > > ### Author Response · Authors · 2018-11-20
> > > > > > > > > > **should work now**
> > > > > > > > > >
> > > > > > > > > > Thank you for your interest in the proof.
> > > > > > > > > >
> > > > > > > > > > It seems like the issue might have been that openreview added the period to the URL.  We edited the earlier comment to remove the period.  Clicking on the link should work now (it seems to us to work in incognito mode).

---

### Official Review · AnonReviewer1 · 2018-11-02
**This is an interesting work with huge potential.**

**Rating:** 8
**Confidence:** 4

**Review:**

The paper is dedicated to computation of singular values of convolutional layers. While singular values of convolutional layers represent sufficient interest for researchers, huge computational complexity made it difficult to investigate their properties in the case of layers of deep neural networks. Using the fact that operator matrix of the convolutional layer has a special form (i.e. can be represented as block-matrix, which blocks are doubly block circulant matrices) the authors proposed a more efficient method of computation of singular values. I really enjoyed reading this paper and I think that it opens a lot of interesting applications. As one of the possible applications the authors proposed a regularization method based on bounding of singular values.

The paper from my point of view has two main drawbacks:

1.  Diversity of experiments. While the paper has strong theoretical component, the part dedicated to experiments is not broad enough. It would be interesting to see regularization on other architectures and other datasets.

2.The system of references. I would recommend to add not only references to the sources, but also to the theorem numbers or the chapters. For example, I would recommend to replace ‘Poposition 9 ((Lefkimmiatis et al., 2013))’ with ‘Poposition 9 ((Lefkimmiatis et al., 2013, Proposition 1))’. In pure math papers, it is a standard rule to add such additional information since many papers contain a lot of theorems and it significantly simplifies reading and understanding the paper.

Despite these disadvantages this is a great work with huge potential.

---

> ### Author Response · Authors · 2018-11-08
> **system of references updated**
>
> Thank you for your review. We have updated the references in the revised version as you requested.

---

### Public Comment · (anonymous) · 2018-10-28
**Questions**

Dear authors,

interesting work. Allow me to phrase few questions:

1) The code in page 2 (numpy) is supposed to return the scalar singular values right? Because it seems to return 3D matrices. Are those different singular values than the typical SVD?

2) In Fig. 4 (Resnet singular values), it seems that only the first layer includes some large singular values. What's the ratio of the singular values, e.g. the first against the rest in every layer? Is it similar across the layers?

3) What does the x axis (measured in seconds) represent in Fig. 5? Test error during training phases?  Regardless of the axis value, it seems that the final error does not improve much if clipping is performed. What's the take of authors on that?

Thanks in advance for your time.

---

> ### Author Response · Authors · 2018-10-29
> **responses to your questions**
>
> Thanks very much for your compliment and your careful reading of our paper.  Here are the answers to your questions.
>
> 1) The singular values are the components of the tensor output by the code on page 2; flattening it produces them in a list.
>
> 2) In Figure 4, while singular values in the first layer are much larger than the rest, many layers have a lot of singular values that are pretty big.  For example, most of the layers have at least 10000 singular values that are at least 1.  As you requested, we have created a plot of the ratios of the singular values in each layer with the largest singular value in that layer.  It can be viewed at https://www.dropbox.com/s/15ujg0qdr9didr8/resnet_svd_ratios.png?dl=0.  The effective rank of the convolutional layers is larger closer to the inputs.
>
> 3) The x-axis in Figure 5 is wall-clock training time using a GPU.  The y-axis is the test error measured while training.  Applying the clipping reduces the test error at convergence from 6.2% to 5.3%, a significant amount.

---

> > ### Author Response · Authors · 2018-10-29
> > **another plot**
> >
> > The first plot that you requested showed that different convolutional layers had significantly different numbers of non-negligible singular values.  We were curious to what extent this was due to the fact that different layers simply were of different sizes, so that the total number of their singular values, tiny or not, was different.  To look into this, instead of plotting the singular value ratios as a function of the rank of the singular values, as in the first plot, we normalized the values on the horizontal axis by dividing by the total number of singular values.  The resulting plot is available at https://www.dropbox.com/s/m0xun9jdc7kd5ry/resnet_svd_percentile_vs_ratio.png?dl=0.

---

> > ### Public Comment · (anonymous) · 2018-11-01
> > **Thanks for the replies**
> >
> > Thanks for the replies.
> >
> > However, question 1 is still not clear. Particularly why your singular values differ from those used in similar works where they reshape the 4D tensor to a 2D matrix.
> > In addition, the some of the singular values with the proposed method seem to be repetitive.

---

> > > ### Author Response · Authors · 2018-11-02
> > > **repeated singular values, and differences with reshaped filter tensors**
> > >
> > > If we have a 4D tensor of dimensions (a, b, i, o) where i, o are the numbers of input and output channels respectively, and (m, n, i) is the size of the input, then the corresponding linear transformation is a matrix of size imn x mno. Thus we would expect min(mni, mno) singular values.  Except in degenerate cases, this linear transformation has full rank.
> > >
> > > In our code, the FFT produces a tensor of dimensions (m, n, i, o) and the second line performs an SVD for the i x o matrix for each value of the first two dimensions. Thus it produces min(i, o) singular values for each on the mn matrices, thus the total number of singular values we produce is mn * min(i, o). So our method produces the correct number of singular values.
> > > (In fact, our unit tests verify that our method computes the correct singular values, as proved in the paper.)
> > >
> > > Singular values are repeated due to the special structure of the linear transformation. This is true even for a 1-D convolution with only one input-output channel. For instance, consider the filter (a, b, c) applied to a 1-d input of 3 pixels. The matrix encoding its linear transformation is ((a b c), (c a b), (b c a)). Its eigenvalues are a+b+c, a + bω + cω^2, a + bω^2 + cω, where ω = (-1 + sqrt(-3))/2. The singular values are the magnitudes of the eigenvalues, but since |a + bω + cω^2| =  |a + bω^2 + cω| we get repeated singular values.  However, for random inputs, we have seen that the repeated singular values tend to come in pairs, so that there are Omega(mn * min(i, o)) distinct singular values.
> > >
> > > It is clear that no reshaping of the 4D tensor of dimensions (a, b, i, o) to a 2D matrix can produce Omega(mn * min(i, o))  singular values.  An example in which the reshaping described in the ICLR’18 paper by Miyato, et al does not compute even the largest singular value correctly can be obtained by executing the following commands after importing numpy as np and defining SingularValues as in page two of our paper:
> > >
> > >    kernel = np.array(np.ones(4)).reshape(2,2,1,1)
> > >    SingularValues(kernel, [4,4])
> > >    reshaped = kernel.reshape(4,1)
> > >    np.linalg.svd(reshaped, compute_uv=False)
> > >
> > > The correct largest singular value of the convolutional layer is 4. (An all-ones feature map is turned into an all-fours feature map by applying this filter.) The (only) singular value of the reshaping of the kernel, which is (1,1,1,1)^T, is 2.  This is not an isolated instance --- random inputs also produce counterexamples except in very rare cases.

---

> > > > ### Public Comment · (anonymous) · 2018-11-04
> > > > **Thanks**
> > > >
> > > > Thanks for your detailed reply.

---

### Public Comment · (anonymous) · 2018-11-06
**Comparison to related work**

Several recent papers have investigated constraining the Lipschitz constant of neural networks as a means to perform some sort of regularisation:

Takeru Miyato, Toshiki Kataoka, Masanori Koyama, and Yuichi Yoshida. Spectral Normalization for Generative Adversarial Networks. In NIPS 2017.

Yusuke Tsuzuku, Issei Sato, and Masashi Sugiyama. Lipschitz-Margin Training: Scalable Certification of Perturbation Invariance for Deep Neural Networks. In NIPS 2018.

Kevin Scaman and Aladin Virmaux. Lipschitz regularity of deep neural networks: analysis and efficient estimation. In NIPS 2018.

Henry Gouk, Eibe Frank, Bernhard Pfahringer, and Michael Cree. Regularisation of Neural Networks by Enforcing Lipschitz Continuity. arXiv preprint arXiv:1804.04368, 2018.

In particular, all of these methods involve using some modification of the power method specialised for convolutional layers. How do these compare with the proposed approach?

---

> ### Author Response · Authors · 2018-11-06
> **comparisons**
>
> We compared with the work on Miyato, et al in our submission.  We have elaborated on this comparison elsewhere in this comment section, including our response to Reviewer 2.  In that response, we demonstrate that there is no reshaping of the kernel tensor whose singular values coincide with the singular values of its linear transform.
>
> In the case of convolutional layers, the paper by Tsuzuku, et al estimates the largest singular value to within a constant factor (see Corollary 1), whereas we characterize the exact values of all of its singular values.
>
> The paper by Scaman and Virmaux uses a power method for individual layers. In the form they describe, it only gives the largest singular value.  We compared our work with the paper by Gouk, et al in our submission.  They also compute an approximation to the largest singular value.  Proposition 8 in our paper (from earlier work) shows that computing the projection of a matrix onto an operator norm ball may require clipping multiple singular values.  The method of Gouk, et al scales down all of the singular values by the amount needed to bring their estimate of the largest to its desired value - this is not closest matrix in the operator norm ball.
>
> Our method is the first to feasibly provide access to all of the singular values of convolutional layers for models commonly used in practice.  It thereby enables study of the properties of trained models, such as whether the linear transformations computed by convolutional layers are essentially full rank.  It also opens the door to a variety of regularizers.

---

> > ### Public Comment · (anonymous) · 2018-11-07
> > **Thanks!**
> >
> > Thanks for the detailed response---It has helped to improve my understanding of how this paper fits in with the related work. Being able to compute the full set of singular values sounds very useful!

---

### Public Comment · (anonymous) · 2018-11-14
**what if we just clip the singular values of the filter coefficients matrix K**

Nice method to calculate the singular values of A. But convolution is just linear transformation, and it is very cheap to calculate the singular values of this linear transformation matrix K. I am curious that why not just clip the singular values of K? Compared with clipping K, what is the benefit of clipping the singular values of the huge circular matrix A? Can the authors compare these two choices in the experiments (clipping K vs A)?

---

> ### Author Response · Authors · 2018-11-20
> **experimental results**
>
> Thank you for your comment.
>
> The largest factor by which a convolutional layer blows up a signal is equal to the largest singular value of the function computed by the layer, which is the largest singular value of A.  We have shown that its singular values, including its largest, are different from those of any reshaping of K.
>
> However based on your suggestion we conducted a few experiments on the CIFAR10 dataset, with somewhat surprising results:
> (1) We clipped the singular values of the reshaped K every 100 steps. We tried various constants for the clipped value, and found that the best accuracy we achieved was the same as the accuracy we achieved by clipping A.
> (2) We clipped the singular values of K to these same values every step, and found that the best accuracy achieved was slightly worse than the accuracy achieved in step 1. So clipping after every step is not very useful. We observed similar behavior when we clipped A after every step.
> (3) Most surprisingly, we found that clipping A on a GPU was about 25% faster than clipping K --- on the same machine, 10000 batches of CIFAR10 took 14490 s when we clipped K, whereas they took 11004 s when we clipped A! One possible explanation is parallelization --- clipping K takes O(m^3 k^2) flops, whereas clipping A does m^2 FFT’s, followed by n^2 m x m SVDs, which takes O(m^3 n^2) flops, but these can be parallelized and completed in as little as O(n^2 log n + m^3) time.
>
> Clearly this is only one dataset, and the results may not generalize to other sets. However it does suggest that finding the full spectrum of the convolutional layer may be no worse than computing heuristic approximations, both in classification accuracy and speed.

---

> > ### Public Comment · (anonymous) · 2018-12-24
> > **Thanks!**
> >
> > Thanks for reporting your experimental results here. Interesting, looks like that norm of A could be bounded by that of K up to a certain scaling constant such that clipping K also works?

---

### Public Comment · ~Jens_Behrmann1 · 2018-12-09
**Circular convolution vs zero-padding (2D)**

Dear Authors,
thank you for this nice analysis and method, I was looking for sth like this for quite a while!

On page 2 you cite (Gray, 2006) for the error when making the approximation of a circular convolution when in fact zero-padding is used, but only in the 1D-case. Did you look into this aspect for 2D-convolution or run experiments?

I observed some differences for 2D-conv when comparing the naive-SVD with your efficient SVD-FFT variant (like spectral norm of 1.59 with naive SVD, 1.66 with SVD-FFT for a random conv-layer with 8 channels, 3x3 kernel, 1 padding, no stride).
Thank you very much!

---

> ### Author Response · Authors · 2018-12-14
> **Circular convolution vs zero-padding (2D)**
>
> Thank you for your interest.
> If the wrap-around (hence circulant form) is replaced by zero-padding, the operator matrix becomes a Toeplitz matrix. The analysis of the error for 2D case is investigated in the paper: “ On the Asymptotic Equivalence of Circulant and Toeplitz Matrices”,  Zhihui Zhu, and Michael B. Wakin, IEEE Transactions on Information Theory, VOL. 63, NO. 5, MAY 2017.
> Here, similar to the 1D case, O(1/n) bound is obtained on the error.
> For practical purposes such as regularizing the operator norm, this small error does not influence the result, while the approximation enables us to find all singular values efficiently.

---

### Meta-Review · Area_Chair1 · 2018-12-13

**Confidence:** 3
**Recommendation:** Accept (Poster)

**Metareview:**

This paper proposes an efficient method to compute the singular values of the linear map represented by a convolutional layer. It makes uses of the special block-matrix form of convolutional layers to construct their more efficient method. Furthermore, it shows that this method can be used to devise new regularization schemes for DNNs. The reviewers did note that the diversity of the experiments could be improved, and R2 raised concerns that the wrong singular values were being computed. The authors should add a section clarifying why the singular values of a convolutional linear map are not found directly by performing SVD on the reshaped kernel - indeed the number of singular values would be wrong. A contrast with the singular values obtained by simple reshaping of the kernel would also be helpful.